# FFCA-NET: STEREO IMAGE COMPRESSION VIA FAST CASCADE ALIGNMENT OF SIDE INFORMATION

## ABSTRACT

Multi-view compression technology, especially Stereo Image Compression (SIC), plays a crucial role in car-mounted cameras and 3D-related applications. Interestingly, the Distributed Source Coding (DSC) theory suggests that efficient data compression of correlated sources can be achieved through independent encoding and joint decoding. This motivates the rapidly developed deep-distributed SIC methods in recent years. However, these approaches neglect the unique characteristics of stereo-imaging tasks and incur high decoding latency. To address this limitation, we propose a **F**eature-based **F**ast **C**ascade **A**lignment network (FFCA-Net) to fully leverage the side information on the decoder. FFCA adopts a coarse-to-fine cascaded alignment approach. In the initial stage, FFCA utilizes a feature domain patch-matching module based on stereo priors. This module reduces redundancy in the search space of trivial matching methods and further mitigates the introduction of noise. In the subsequent stage, we utilize an hourglass-based sparse stereo refinement network to further align inter-image features with a reduced computational cost. Furthermore, we have devised a lightweight yet high-performance feature fusion network, called a Fast Feature Fusion network (FFF), to decode the aligned features. Experimental results on InStereo2K, KITTI, and Cityscapes datasets demonstrate the significant superiority of our approach over traditional and learning-based SIC methods. In particular, our approach achieves significant gains in terms of 3 to 10-fold faster decoding speed than other methods.

## 1 INTRODUCTION

With the rapid advancement of stereoscopic imaging technologies and the increasing popularity of binocular imaging devices, stereo images have found wide applications in crucial fields such as autonomous driving, augmented reality Livatino et al. (2012), video surveillance, and robot navigation Murray & Little (2000). As a result, there is an urgent requirement to efficiently process and transmit massive amounts of stereo image data. For example, a vehicle equipped with binocular cameras for autonomous driving generates approximately 1GB of data per second. Hence, the development of effective stereo image compression techniques has become increasingly significant.

Unlike single-image compression, Stereo Image Compression (SIC) not only focuses on reducing redundancy within each image but also considers the correlation between images captured from different viewpoints to achieve higher coding efficiency. In general, most deep learning methods follow existing multi-view coding standards, such as H.265-based MV-HEVC. Tech et al. (2015) employing a joint encoding structure to compress images from different viewpoints. These approaches first compress the auxiliary views of stereo images using single-image compression methods. Then, during the compression of the main view, redundant information between stereo images is eliminated through disparity-compensated prediction, and only the residual after prediction needs to be encoded. Thanks to advancements in deep single-image compression algorithms Ballé et al. (2016; 2018) and stereo-matching techniques, recent developments in stereo-image compression have benefited greatly. Some works adopt traditional one-way encoding techniques, such as Liu et al. (2019), Deng et al. (2021), and Wödlinger et al. (2022). These approaches follow a strict sequential encoding order, propagating potential representations of auxiliary views as context into the encoding branch of the main view and employing disparity estimation or depth homography estimation to remove redundancy. Additionally, Lei et al. (2022) introduces a novel context dependency between views, compressing binocular images and extending the one-way encoding mechanism to bidirec-

tional encoding. These works demonstrate the significant improvement in compression efficiency achieved by deep learning methods in SIC scenarios. However, the encoders of these methods tend to be excessively large. In practical applications of stereo images, such as in-car cameras and VR devices, the terminal encoders lack powerful computational capabilities, making it more suitable to perform complex computations at decoder terminals, such as cloud servers.

According to the theory of Distributed Source Coding (DSC) Slepian & Wolf (1973); Wolf (1973); Wyner & Ziv (1976), encoding correlated data sources independently and utilizing side information at the decoder can achieve the same compression rate as joint encoding. In recent years, there have been some proposed deep learning algorithms based on distributed coding frameworks. In attempts to achieve this asymmetric structure, integration of side information at the decoding stage was explored in Mital et al. (2022) and Ayzik & Avidan (2020). However, effective alignment between different sources of information was not achieved. On the other hand, Huang et al. (2023) and Zhang et al. (2023) utilized complicated patch mapping and attention modules, respectively, in the feature domain to capture contextual information between images. These methods failed to fully exploit the priors provided by the stereoscopic image scene, resulting in unsatisfactory decoding speed.

To effectively incorporate side information at the decoder in SIC, this paper proposes a Feature-based Fast Cascade Alignment network (FFCA). The main idea of our proposed course-to-fine cascade structure is to perform coarse-grained matching of features using a priori-based stereo patch-matching module in the feature domain. We then employ an hourglass-like stereo rectification network to achieve fine-grained alignment in a sparse feature space. The aligned feature information is fed into a fast feature fusion layer (FFF) for image reconstruction. Compared to state-of-the-art SIC compression algorithms, our method achieves higher-quality reconstructed images with lower bit consumption and significantly faster decoding speed, ranging from several to tens of times faster.

The main contributions of this paper can be summarized as follows:

- We propose a stereo patch matching technique that utilizes features and prior knowledge of stereo images to achieve more precise alignment at the decoding end.

- We develop a pyramid-based sparse stereo refinement network and a lightweight feature fusion module to efficiently refine the matched features obtained from stereo patch matching and effectively fuse the aligned features for reconstructed images.

- We conduct extensive experiments on three large-scale high-resolution stereo datasets to validate the outstanding performance of our method in SIC. Additionally, our approach demonstrates significantly faster decoding speed compared to existing learning-based methods.

## 2 RELATED WORK

The intent of image compression algorithms is to explicitly or implicitly construct a superior representation compared to the original image space. Traditional image compression methods often rely on manually crafted transform representations, such as discrete cosine transform or inter-block prediction after image partitioning Wallace (1992); Taubman et al. (2002). On the other hand, learning-based end-to-end image compression algorithms Li et al. (2018); Minnen et al. (2018); He et al. (2021); Mentzer et al. (2018) attempt to seek a more compressed representation through a trainable non-linear transformation.

**Learned Single Image Compression** The development of deep neural networks has propelled the introduction of deep compression algorithms. Ballé et al. (2016) pioneered an end-to-end model based on autoencoders with rate-distortion loss as the optimization objective. Subsequently, this work has been expanded upon: Ballé et al. (2018) . introduced spatial adaptation, factorization, and hyperprior entropy models. Minnen et al. (2018) incorporated autoregressive context modeling into the prior, significantly improving performance at the cost of decoding complexity. Cheng et al. (2020). proposed a more accurate modeling of the latent distribution using discrete mixture models, while He et al. (2021) proposed a parallel chessboard-style context model to speed up decoding.

**Stereo Image Compression** Stereo image compression requires considering the correlation between views to save more bitrate. There are many works on learning-based stereo image compression, with

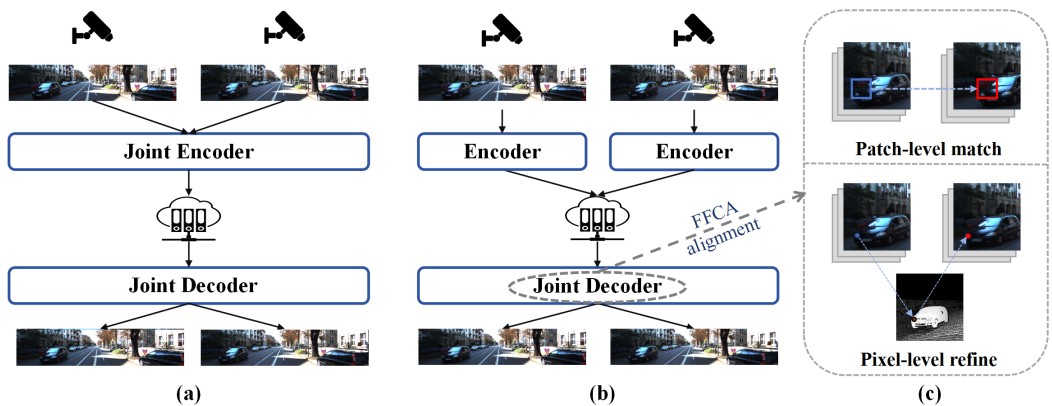

Figure 1: Overview of various structures for stereo image coding, including (a) joint encoding architecture and (b) asymmetric DSC structure. (c) briefly outlines the coarse-to-fine alignment method employed in our proposed FFCA-Net.

most of them following a single-sided encoding approach. This means that the auxiliary image is independently encoded, and its contextual information is fused into the main image for encoding. For example, Liu et al. (2019) uses a neural network in the feature domain to estimate disparity and incorporates aligned auxiliary image context through skip modules. Deng et al. (2021) employs a deep homography estimator to fit the correlation in stereo images and utilizes a high-performance GMM-based context entropy encoder to estimate residual after prediction. Wödlinger et al. (2022) learns element-wise shifts between viewpoints through an encoder optimized with MSE. Lei et al. (2022) explores the possibility of bidirectional encoding, utilizing bidirectional contextual transformation modules and bidirectional conditional entropy models, achieving additional bitrate savings for both views after compression. However, the encoders of these algorithms tend to be complex in order to incorporate inter-image information, and the decoders often prioritize pixel-level prediction and alignment, resulting in suboptimal decoding speeds.

**Learned Distributed Source Coding** Indeed, there are relatively few works on learning-based distributed coding. Ayzik & Avidan (2020) proposed using patch matching in the image domain to reconstruct higher-quality images by exploiting a large amount of similarity or overlap between different views. However, this matching lacks robustness and exhibits suboptimal performance. Zhang et al. (2023) employed a cross-attention mechanism to capture global correlations among different viewpoints, surpassing the compression performance of joint encoding-decoding frameworks. However, in order to provide the decoding end with side information, this method necessitates additional design modifications to the encoder to meet the requirement. Mital et al. (2022) used a feature extractor to extract features of side information and combined it with the main information for auxiliary decoding. Nevertheless, this method did not consider registration between views, and the results tend to be less satisfactory when there is a significant disparity between the views captured by the cameras. To rectify this deficiency, Huang et al. (2023) proposed a patch-matching approach in the multi-scale feature domain, enabling a more effective fusion of side information and yielding astonishing encoding benefits. Although these methods are designed only at the decoding end, they fail to fully consider the inherent relationship between stereo images, leaving room for optimization in the task of stereo image compression.

## 3 METHODOLOGY

FFCA employs a cascaded structure that operates in a coarse-to-fine manner, facilitating swift and efficient alignment between feature layers of disparate perspective views. In specific terms, FFCA can be divided into two components: stereo patch matching and hourglass-based sparse stereo refinement. Figure 2 delineates the architectural framework of our method: our primary view image is initially directed into a baseline single-image encoder-decoder, yielding a range of multi-scale primary view features denoted as $\boldsymbol{h}_{\hat{x}}^i$ are extracted from the decoder of the upsampling structure. Simultaneously, auxiliary view features denoted as $\boldsymbol{h}_{\hat{y}}^i$. Here, $i$ signifies that the layer represents the

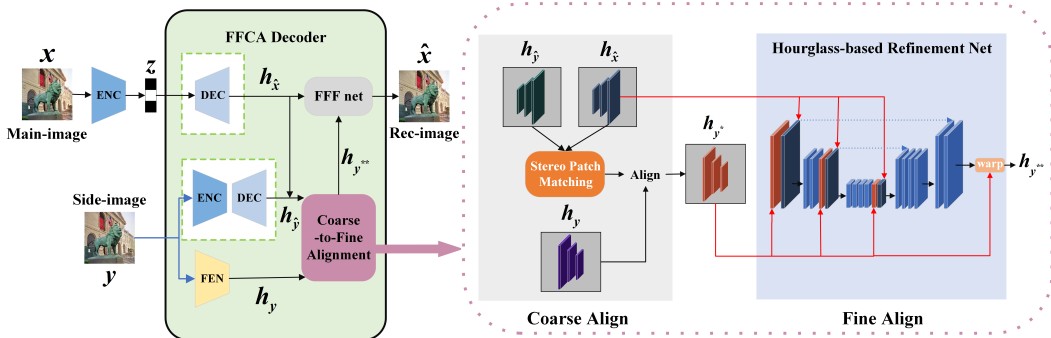

Figure 2: The overview of the proposed model architecture. ENC and DEC refer to the encoder and decoder of the baseline single-image compressor, respectively. FEN represents the feature extraction network used to extract precise side information features.

feature map obtained after the $i$-th iteration of upsampling with a scale $= 2$ in the decoder, using the latent code as input. Compared with the MSFDPM method (Huang et al. (2023)), we have employed a more lightweight feature extractor to capture multi-scale lossless side information.

## 3.1 STEREO PATCH MATCHING ON MULTI-SALE FEATURE-DOMAIN

We have observed that stereo images exhibit a fixed direction of horizontal displacement for rigid transformations in the image domain, a characteristic that is also preserved in the features extracted by general CNN-based models. In fact, this has been confirmed by many works in the field of SIC. Our proposed stereo patch matching technique is based on this super-prior. Subsequently, for a given $i$, we perform sampling on $\boldsymbol{h}_{\hat{x}}^i$ with a window size of $B$. The strides of the window sliding are set to $S$. Once all the sampling is completed, we define the collection of patches obtained from all the sampled windows as:

$$\mathcal{P}\left(\boldsymbol{h}_{\hat{x}}^i, B, S\right) = \left\{p\left(\boldsymbol{h}_{\hat{x}}^i, B, S, m, n\right)\right\}, \text{ where } m = 0, \cdots, \left\lfloor \frac{H-B}{S} \right\rfloor, n = 0, \cdots, \left\lfloor \frac{W-B}{S} \right\rfloor. \quad (1)$$

Here, $\mathcal{P}$ represents the set of the overall sampling, while $p$ denotes a specific sampled patch within it, with $m, n$ representing the coordinates of that patch. Based on this definition, we sample a set $\mathcal{P}\left(\boldsymbol{h}_{\hat{x}}^i, B, B\right)$ from $\boldsymbol{h}_{\hat{x}}^i$. It is important to note that there is no overlap between each patch in this set. For each patch in the above set, we aim to find the most similar window in $\boldsymbol{h}_{\hat{y}}^i$ that closely resembles it. To accomplish this objective, we similarly sample $\mathcal{P}\left(\boldsymbol{h}_{\hat{y}}^i, B, 1\right)$. Actually, when the size of $\boldsymbol{h}_{\hat{y}}^i$ is large, the resulting patch collection $\mathcal{P}$ sampled from it will be exceedingly vast. This leads to lower algorithm efficiency and an increased likelihood of erroneous matches. To address this, we leverage the prior knowledge of stereo images to narrow down the matching range. For each patch from $\mathcal{P}\left(\boldsymbol{h}_{\hat{x}}^i, B, B\right)$ we restrict our search in the $\boldsymbol{h}_{\hat{y}}^i$ to windows located in the same row as the patch block and within the disparity direction, defined as $\vec{\mathcal{P}}_m\left(\boldsymbol{h}_{\hat{y}}^i, B, 1\right)$. Subsequently, we can calculate the distance between the target patch and this search set:

$$\rho\left(p\left(\boldsymbol{h}_{\hat{x}}^i, B, B, m, n\right), \vec{\mathcal{P}}_m\left(\boldsymbol{h}_{\hat{y}}^i, B, 1\right)\right). \quad (2)$$

Here $\rho(\cdot, \cdot)$ refers to the cosine distance, where a smaller distance indicates a higher similarity between two patches. The computation of this distance is equivalent to seeking the most similar patch within the search range to the target patch. For the sake of simplicity, *we denote the aforementioned distance as $\rho_{m,n}$*. This super-prior is reasonable, as illustrated in the Figure 3. Although adopting a greedy search strategy expands the search space multiple times, it often leads to incorrect matching when dealing with dissimilar patches that exhibit significant positional differences across different

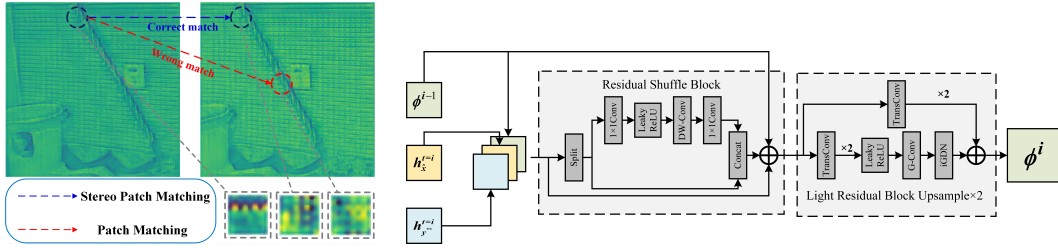

Figure 3: Different match results.    Figure 4: One iteration of fast feature fusion network.

viewpoints. On the other hand, stereo patch matching consistently manages to find the correct patch pairs under the same circumstances.

It is worth noting that due to the constraint on the search space for patch matching, we can proceed with parallel searching for patches from set $\mathcal{P}\left(\boldsymbol{h}_{\hat{x}}^i, B, B\right)$ that are located on different rows. To accomplish this, we have devised a grouped convolution approach that enables parallel computation of correlation coefficients, resulting in a significant speed boost for the matching process.

Next, we establish the mapping relationship for all $m, n$:

$$u(m,n), v(m,n) = \left\{u, v \mid \rho\left(p\left(\boldsymbol{h}_{\hat{y}}^i, B, 1, u, v\right), p\left(\boldsymbol{h}_{\hat{x}}^i, B, B, m, n\right)\right) = \rho_{m,n}\right\}. \quad (3)$$

Based on the extracted lossless side information $\boldsymbol{h}_y^i$, we can rearrange the information into patches to obtain $\boldsymbol{h}_{y^\star}^i$ using the aforementioned mapping:

$$p\left(\boldsymbol{h}_{y^\star}^i, B, B, m, n\right) = p\left(\boldsymbol{h}_y^i, B, 1, u(m,n), v(m,n)\right). \quad (4)$$

Indeed, patch matching on feature layers at every scale is a highly complex and unnecessary endeavor, as it inadvertently introduces superfluous noise Huang et al. (2023). Inspired by this work, we employed the approach of *Reusing First Feature Layer Inter-Patch Correlation*. This method involves performing patch matching solely in the high-resolution feature layer at $i = 1$. The obtained $u(m,n)$ and $v(m,n)$ from the matching process will serve as guidance, with corresponding scaling, for aligning the remaining feature layers. Specifically, we restrict the stereo-patch matching to only occur at $i = 1$, where we compute the inter-patch correlation and obtain the mapping relationships by 3 to obtain $u^1(m,n), v^1(m,n)$. During the matching process in the remaining layers $\{i = 2, 3, 4\}$, we maintain these inter-patch mapping relationships. However, due to the dimensional variations in these layers, we need to apply corresponding transformations to the indices of the mappings:

$$u^i(m,n), v^i(m,n) = 2^{i-1} * u^1(m,n), 2^{i-1} * v^1(m,n). \quad (5)$$

## 3.2 HOURGLASS-BASED SPARSE STEREO REFINEMENT

Numerous studies in stereo matching Shen et al. (2021); Gu et al. (2020); Zhou et al. (2020); Chang & Chen (2018) have emphasized the importance of utilizing multi-scale features. However, these approaches often rely on a wide range of disparity searches and the construction of 3D convolutions, resulting in high computational costs. To efficiently perform alignment in the feature domain, we propose a sparse stereo rectification network in an hourglass-style architecture. The network structure is illustrated in the figure, and more detailed parameters can be found in the appendix. Firstly, we construct a cost volume at different scales:

$$V_{\text{concat}}\left(x, y^\star\right) = \boldsymbol{h}_{\hat{x}} \| \boldsymbol{h}_{y^\star}. \quad (6)$$

Here, $\|$ denotes the operation of concatenation along the channel dimension. Since low-resolution feature layers do not provide accurate disparity information, we exclude the lowest-resolution features (i.e., $i = 4$) from the operation. To reduce computational complexity, we employ grouped convolution layers with skip connections to regularize and fuse features at different scales. Additionally, a grouped convolution module with a downsampling structure is utilized to downsample the fused features at the highest resolution, which are then merged with the features of the next scale.

Once all feature volumes are connected to the encoder, we apply grouped transposed convolution to perform upsampling. The network's output is $dp_1$, a 2D disparity map of size $D \times H_1 \times W_1$, where $H_1, W_1$ represent the height and width of the $\boldsymbol{h}_{\hat{x}}^i$, and $D$ represents the disparity range. We will acquire $\{dp_i, i = 2, 3, 4\}$ through downsampling of $dp_1$. Due to the purpose of this model, which is to perform fine-grained refinement after stereo patch matching, we only need to set a smaller disparity search range, significantly increasing the efficiency of the network.

However, applying pixel-level disparity uniformly across all feature channels may not be an optimal strategy. Based on empirical observations, we have found that the variations in features between the main information and the side information are non-uniform across channels. The distribution of these differences tends to follow a long-tail distribution, where a few channels exhibit significantly larger differences compared to the rest. This implies that different channels require varying degrees of alignment accuracy. In stereo images, there are numerous structurally similar features, and their corresponding channels may not require additional alignment. To address this challenge, we propose a sparse alignment strategy. we actively select a subset of channels with significant differences while freezing the remaining channels, allowing the disparity map to only affect these selected channels. This approach prevents the introduction of unnecessary noise from channels with smaller differences during training and avoids overcorrection on these channels, which could hinder subsequent decoding processes. Based on this observation, we can define channels that exhibit significant differences:

$$G = \left\{ g \mid \| \boldsymbol{h}_{\hat{x};g}^i - \boldsymbol{h}_{y^\star;g}^i \|_2 \geq \mu \right\}, \tag{7}$$

where $\boldsymbol{h}_{\odot;g}^i$ represents the $g$-th channel of the feature volume $\boldsymbol{h}_{\odot}^i$, and $\mu$ is a hyperparameter. Here, $G^c$ refers to the complement of $G$, representing the set of feature channels that are not selected. Then, we perform warp operations using the 2D disparity map only on these selected channels. Finally, we have obtained the side information features $\boldsymbol{h}_{y^{\star\star}}^i$ after performing coarse-to-fine matching, where:

$$\boldsymbol{h}_{y^{\star\star};g}^i = \left\{ \begin{array}{ll} \mathrm{Warp}(\boldsymbol{h}_{y^\star;g}^i, dp_i), & g \in G \\ \boldsymbol{h}_{y^\star;g}^i, & g \in G^c \end{array} \right. . \tag{8}$$

To efficiently and rapidly integrate feature blocks $\boldsymbol{h}_{\hat{x}}$ and $\boldsymbol{h}_{y^{\star\star}}$, we have devised the Fast Feature Fusion (FFF) network, as shown in Figure 4. The structure of FFF follows a similar pattern as in Huang et al. (2023). Taking inspiration from Zhang et al. (2018), we employ a network that utilizes shuffle blocks and depthwise separable convolutions. At $i$-th stage ($i = 1, 2, 3, 4$) of the FFF, the input consists of the aligned feature block $\boldsymbol{h}_{\hat{x}}^i, \boldsymbol{h}_{y^{\star\star}}^i$ and output from the previous stage, defined as $\phi^{i-1}$. The input is first passed through a shuffle block to fuse features and then undergoes a lightweight upsampling block to output a higher-resolution feature block. The final output of the network is obtained by adding it to the reconstructed image from a single-image decoder.

### 3.3 Loss Fuction

The training problem of the FFCA model is equivalent to a joint optimization problem of compression rate and distortion. Simultaneously, we aspire for our pixel-level refinement network to converge, necessitating the inclusion of inter-view feature distortion to aid in training. Hence, a training loss composed of three metrics is used:

$$\mathcal{L} = R(\hat{\boldsymbol{z}}) + \lambda \left( (1 - \alpha)d_1\left(\boldsymbol{x}, \hat{\boldsymbol{x}}\right) + \alpha d_2\left(\boldsymbol{h}_{\hat{x}}^1, \boldsymbol{h}_{y^\star}^1\right) \right). \tag{9}$$

Here, $d_1(\cdot, \cdot)$ refers to the reconstruction loss between $\boldsymbol{x}$ and $\hat{\boldsymbol{x}}$, while $d_2(\cdot, \cdot)$ represents the distortion between the main image feature block and the side information feature block. $R(\cdot)$ denotes the compression rate of the latent representation $\boldsymbol{z}$. $\lambda$ is the weight that controls the trade-off between distortion and compression rate, while $\alpha$ is the weight that balances the two types of distortion.

## 4 Experiments

### 4.1 Experimental Setup

**Datasets.** We validate our method on three high-resolution stereo image datasets: KITTI-stereo Menze & Geiger (2015), Cityscapes Cordts et al. (2016), and InStereo2K Bao et al. (2020). KITTI-stereo and Cityscapes represent outdoor distant views, while InStereo2K represents indoor near views.

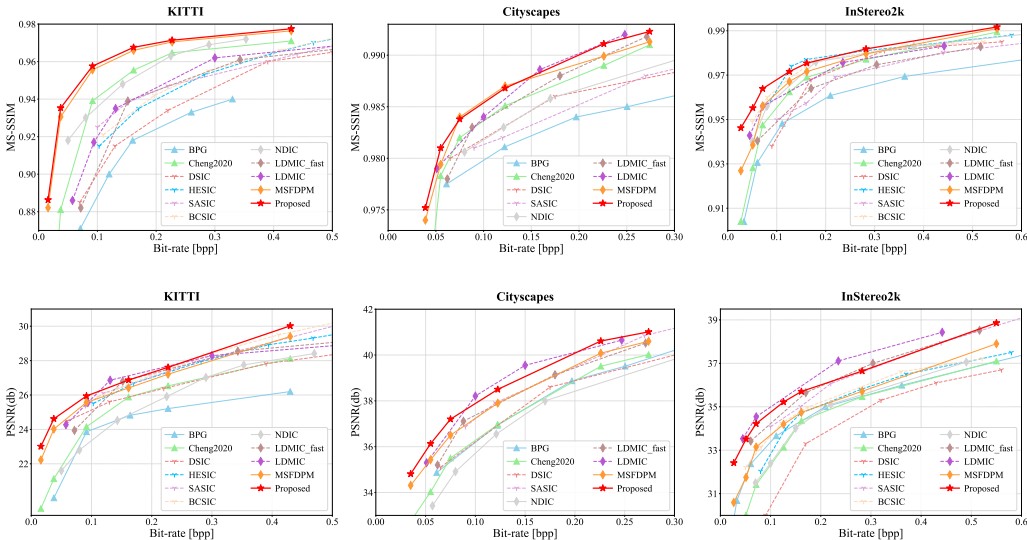

Figure 5: Rate–distortion curves for PSNR (dB) and MS-SSIM with various compression methods.

**Metrics.** Bits per pixel (bpp) is used to measure the bitrate. For assessing image quality, peak signal-to-noise ratio (PSNR) and multi-scale structural similarity (MS-SSIM) Wang et al. (2003) are utilized. These two metrics are widely recognized for evaluating distortion in image reconstruction. Additionally, we apply Bjøntegaard delta PSNR (BD-PSNR) Bjontegaard (2001) to evaluate bitrate savings at the same level of distortion, and BD-rate to determine PSNR gainings at the same level of bitrate.

**Baseline.** We compare three categories of baseline models: (1) *Single-image compression models*: This includes the traditional algorithm BPG Bellard (2014) and the learning-based method Cheng et al. (2020). Specifically, we employ the version of "cheng2020" implemented by Bégaint et al. (2020). (2) *Joint encoding-decoding stereo image compression models*: This encompasses HESIC Deng et al. (2021), SASIC Wödlinger et al. (2022), BCSIC Lei et al. (2022), and DSIC Liu et al. (2019) mentioned earlier. Among these, for HESIC and BCSIC, we used the results reported in their respective papers. It should be noted that HESIC and BCSIC have not been validated on the Cityscapes dataset. (3) *Learning-based distributed compression models*, which include NDIC Mital et al. (2022), MSFDPM Huang et al. (2023), and LDMIC(LDMIC-fast) Zhang et al. (2023). Excluding HESIC and BCSIC, we re-evaluated the rest of the baseline models utilizing their open-source codes and published parameters. For the LDMIC model's evaluation, to ensure a fair comparison, we abstained from the fine-tuning strategy mentioned in Zhang et al. (2023).

**Implementation Details** Our proposed method is implemented using PyTorch Paszke et al. (2019). Experiments were conducted on two Intel(R) Xeon(R) Silver 4210 CPUs and two NVIDIA 2080ti GPUs. The Adam optimizer Kingma & Ba (2014) was employed with a learning rate of $1 \times 10^{-4}$. Other hyper-parameters include: **(i)** The hyper-parameter for filtering significant inter-feature channels, with $\mu = 0.5$. **(ii)** The patch size set at $B = 16$. **(iii)** The weight for two stages of distortions, defined as $\alpha = 0.1$. For more experimental details, please refer to Appendix 6.2.

## 4.2 RESULTS AND ANALYSIS

**Quantitative results.** Table 1 presents the BD-rate results of our method and other approaches, using BPG as the baseline. A lower BD-rate indicates a more significant performance improvement relative to the baseline model. Figure 5 illustrates the RD curves for all compared methods. As mentioned earlier, our approach optimizes based on MS-SSIM, so we evaluated MS-SSIM across all datasets. To maintain consistency with prior works, we also assessed PSNR. Our MSSSIM-based BD-rate outperforms other methods across all datasets. Even when evaluated using PSNR as a criterion, our method surpasses most baseline models.

Table 1: BD-rate comparisons relative to BPG on different datasets, with the best results inred and second-best ones in blue.

| Classifications | Methods | Kitti | | Cityscapes | | InStereo2K | |
|---|---|---|---|---|---|---|---|
| | | PSNR | MS-SSIM | PSNR | MS-SSIM | PSNR | MS-SSIM |
| Single | Cheng2020 | -21.61% | -59.11% | -2.75% | -43.54% | 38.02% | -30.29% |
| Joint | HESIC | -65.98% | -35.13% | - | - | -12.83% | -66.91% |
| | DSIC | -55.33% | -18.64% | -6.89% | -38.67% | 85.37% | -31.98% |
| | SASIC | -68.62% | -50.95% | -23.30% | -21.14% | -34.99% | -26.33% |
| | BCSIC | -69.82% | -40.05% | - | - | -15.96% | -62.14% |
| Distributed | NDIC | 2.83% | -66.42% | 10.02% | -33.15% | 15.24% | -55.21% |
| | MSFDPM | -65.92% | -83.41% | -24.29% | -53.52% | -10.18% | -50.82% |
| | LDMIC-fast | -54.66% | -37.10% | -22.80% | -42.82 % | -41.61% | -31.99% |
| | LDMIC | -63.29% | -43.60% | -38.09% | -49.05% | -58.45% | -55.69% |
| | FFCA(Proposed) | -74.62% | -85.18% | -37.84% | -55.36% | -47.02% | -69.75% |

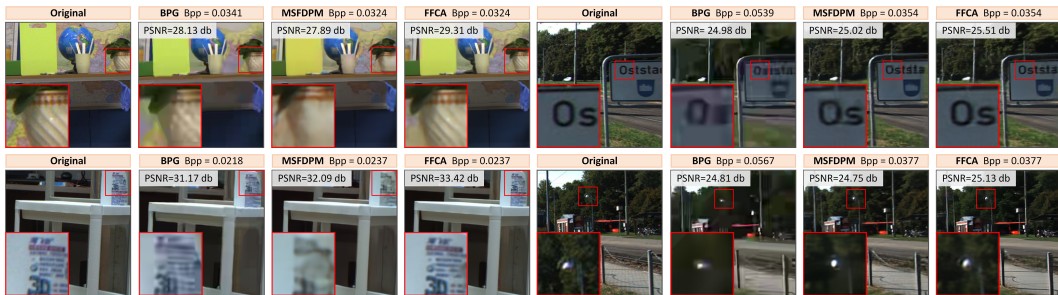

Figure 6: Visual comparison of the reconstructed using our proposed FFCA and the comparison methods including BPG (Bellard (2014)) and MSFDPM (Huang et al. (2023)).

Our method, termed FFCA, demonstrates significant improvements in compression performance when compared to the baseline model. Particularly on the InStereo2K dataset, FFCA achieves an impressive bit savings of 85.04% when evaluated in terms of PSNR. When benchmarked against the joint encoding-decoding schemes, FFCA consistently delivers superior PSNR and MS-SSIM values than these baseline models at comparable bit rates. For instance, when pitted against MSE-optimized algorithms like DSIC (SASIC), FFCA exhibits a substantial reduction in bits across multiple datasets, as quantified by PSNR. When contrasted with the asymmetric DSC baseline, our approach stands out with clear advantages. As previously discussed in Section 3.1, MSFDPM tends to underperform on close-range indoor views, often resulting in mismatched patches. Our innovative stereo-patch matching technique successfully mitigates this problem, leading to substantial bit savings on the InStereo2K dataset, both in terms of PSNR and MS-SSIM. LDMIC, with its integration of multi-head attention modules, sets a high benchmark in compression, especially when assessed using the PSNR metric. Notably, FFCA's performance is nearly on par with LDMIC across various datasets and even surpasses it on the KITTI dataset. Moreover, when judged based on the MS-SSIM metric, our method consistently outshines LDMIC. An additional point worth highlighting is that the computational complexity of FFCA is only comparable to the streamlined version, LDMIC-fast.

**Visualization.** To showcase the compression results, we provide visualizations in Figure 6. For a fair comparison, we ensured similar compression rates across different schemes. Our method achieves higher PSNR values with fewer or equivalent bits compared to traditional approaches like BPG and the deep DSC method MSFDPM. Our algorithm preserves strong structural similarity, even at very low bit rates, avoiding the prominent distortions and artifacts observable in BPG. In comparison to MSFDPM, our fine-grained calibration retains more image details, capturing small text and object textures even at reduced bit rates.

**Computational complexity.** Table 2 compares the FLOPs and decoding latency of our model with baseline models. Owing to the unique structure of asymmetric DSC, it allows for lightweight encoders and parallel encoding, advantages not present in joint encoding-decoding mode. For fairness, we focus on comparing the complexity of decoding. FFCA not only exhibits the lowest FLOPs and decoding latency among all baseline methods but also achieves decoding latency that is 3.06-5.82 times faster when compared to joint decoding methods, and 1.15-4.91 times faster against asymmetric DSC methods. The method MSFDPM (Huang et al. (2023)) shows a decrease in decoding speed due to its greedy strategy-based patch matching, while our stereo-based patch matching achieves a 10-20 times speedup.

Table 2: Computation complexity tested on InStereo2K with the resolution as $832 \times 1024$

| Methods | FLOPs | Time |
|---|---|---|
| DSIC | 3378.65G | 15.03s |
| HESIC | 1122.87G | 28.56s |
| SASIC | 2532.87G | 19.58s |
| NDIC | 1245.89G | 5.64s |
| MSFDPM | 1604.74G | 23.85s |
| LDMIC-fast | 1851.69G | 6.66s |
| LDMIC | 1838.42G | 27.77s |
| FSCA(Propsed) | **781.76G** | **4.91s** |

### 4.3 ABLATION STUDY.

We conducted ablation experiments on the InStereo2K dataset and calculated the BD-rate and BD-PSNR, as shown in Table 3. For the ablation experiments regarding decoding speed, please refer to the appendix for more details.

**Hourglass-based sparse stereo refinement**: The performance of our model without the fine-grained refinement module is represented by "W/O HSSR". As can be observed, omitting this module results in a decrease of approximately 0.23dB at the same bit rate, indicating the effectiveness of this module.

**Stereo patch matching**: "W/O SPM & HSSR" represents our model's performance without both the coarse and fine-grained alignment. Compared to "W/O HSSR", the absence of the Stereo patch matc-

Table 3: Comparison in ablation study

| Model | BD-rate | BD-PSNR |
|---|---|---|
| W/O SPM HSSR | -16.61% | 0.52dB |
| W/O HSSR | -49.31% | 2.04dB |
| W/O FFF | -54.71% | 2.25dB |
| Proposed | -54.51% | 2.27dB |

hing module causes a notable performance drop, with a decrease in BD-PSNR by 1.75 dB. This emphasizes the significance of coarse matching in the initial stage, suggesting that decoding without matching fails to effectively utilize inter-view information.

**Fast Feature Fusion**: The Fast Feature Fusion module is primarily designed to accelerate decoding. However, in our experiments, we found that at lower bit rates, the lightweight decoder slightly outperforms the decoder with a more complex structure. Although a minor performance decline is noticed at higher bit rates, overall, this result validates our adoption of FFF for achieving faster decoding latency.

## 5 CONCLUSIONS

This paper introduces FFCA-Net, a fast cascaded framework for distributed compression of stereo images. Our approach utilizes coarse-to-fine feature matching to align side information features with the main information. Experimental evidence demonstrates that FFCA effectively leverages stereo view information, achieving superior encoding gains while maintaining a significantly lower decoding latency compared to existing methods. Based on this framework, future work can be extended in two aspects. Firstly, extracting more general priors can broaden the applicability of this method to various scenarios. Secondly, exploring more efficient ways to apply these priors in order to accelerate the encoding and decoding processes is worth investigating.

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

# 6 APPENDIX

## 6.1 EXPERIMENTAL DETAILS

### 6.1.1 DATASETS

We have validated our method on three high-resolution stereo image datasets, namely Cityscape Cordts et al. (2016) and Kitti-stereo Menze & Geiger (2015), which represent outdoor distant views, as well as InStereo2K Bao et al. (2020), which represents indoor near views. Cityscape consists of 5000 pairs of $2048 \times 1024$ images, with 2975 pairs for training, 500 pairs for validation, and 1525 pairs for testing. Kitti-stereo comprises 1578 training image pairs and 790 test image pairs, all with the size of $1242 \times 375$. InStereo2K includes 2010 training image pairs and 50 test image pairs, all with a size of $1080 \times 860$.

### 6.1.2 EXPERIMENTAL SETTING

Initially, we trained a single-image compression baseline Cheng et al. (2020). Subsequently, we trained the complete model, where the parameters of the autoencoder were initialized using the pretrained baseline. For the InStereo2K dataset, training results were reported for seven different values of $\lambda$: $\lambda \in \{1, 0.2, 0.1, 0.07, 0.035, 0.02, 0.01\}$. On the KITTI and Cityscapes datasets, results were provided for six different $\lambda$ values: $\lambda \in \{0.5, 0.1, 0.07, 0.035, 0.01, 0.005\}$. The training epochs for the KITTI and Instereo2K datasets were set at 80, while for the Cityscapes dataset, it was set at 100. Across all datasets, a batch size of 16 was used. During the training process, the datasets of KITTI and Instero2K are randomly cropped into blocks of size $320 \times 960$ and $512 \times 512$, respectively, while Cityscape follows the conventional preprocessing approach: for every image, we crop 64, 256, and 128 pixels from the top, bottom, and sides, respectively, to remove the car hood Wödlinger et al. (2022); Zhang et al. (2023). During testing, we employ replication-padding to extend the edges of the feature maps Huang et al. (2023) until the length of the feature maps can be evenly divided by the patch size. After the completion of matching, we will trim the feature maps back to their original size.

## 6.2 ABLATION FOR ACCELERATION

In this section, we will delve into our specific contributions in model acceleration and lightweight design. Our model consists of three components: coarse-grained stereo patch matching, fine-grained module hourglass-based sparse stero refinement and a fast feature fusion module. For each component, we have carefully selected comparable and compelling baselines for comparison.

**Stereo Patch Matching** We have chosen Multi-scale Patch-matching Huang et al. (2023) as our baseline, which is similar to our approach as it also involves coarse-grained matching based on feature level. Our input image size is $832 \times 1024$, resulting in feature map dimensions of $128 \times 416 \times 512$. We conducted inference speed tests for both methods on CPU and GPU, as shown in Table 4. It is evident from the results that our algorithm outperforms the baseline method by nearly

Table 4: Acceleration evaluation of Stereo Patch Matching.

| Method | Inference Speed(CPU) | Inference Speed(GPU) |
|---|---|---|
| Stereo PM (Proposed) | **0.76**s | **0.027**s |
| Multi-scale PM (Huang et al. (2023)) | 15.32s | 0.46s |

20-fold, both in CPU and GPU environments. This significant speed improvement is attributed to our efficient parallel computing techniques, which have proven to be reliable.

**Fast Feature Fusion** We have chosen Feature Fusion Huang et al. (2023) as our baseline. Our proposed FFF module is an enhanced version of the Feature Fusion module, with a smaller parameter

count and faster inference speed. Here, we provide a more detailed explanation of the input and output of the FFF module. For each iteration $FFF^i$, where $i = 1, 2, 3$, it can be abstracted as the following equation.

$$\phi^i = FFF^i(\phi^{i+1}, h_{\hat{x}}^i, h_{y^{\star\star}}^i) \quad i = 1, 2, 3.$$

Since the FFF module cannot access features from the previous layer when fusing the lowest-resolution feature map ($i = 4$), the abstraction of the FFF module at this stage is as:

$$\phi^4 = FFF^4(h_{\hat{x}}^4, h_{y^{\star\star}}^4).$$

Next, we validate our proposed method and the baseline approach on a scene with an input image size of $832 \times 1024$. Table 5 presents the runtime and model parameter count for our method and the baseline method on CPU. The results confirm the effectiveness of our model.

Table 5: Acceleration evaluation of Fast Feature Fusion.

| Method | Inference Speed(CPU) | Parameters |
|---|---|---|
| Fast Feature Fusion (Proposed) | **1.84**s | **3.04**M |
| Feature Fusion (Huang et al. (2023)) | 2.20s | 7.02M |

**Hourglass-based Sparse Stereo Refinement** To the best of our knowledge, the only prior learning-based SIC work that utilizes stereo matching to eliminate inter-view redundancy is DSIC Liu et al. (2019). For fairness, we have chosen the Parametric Skip Function, a crucial component of DSIC, as the baseline method. We conducted validation on a scene with an input image size of $832 \times 1024$. Table 6 presents the runtime and model parameter count for our proposed method and the DSIC baseline on CPU.

Table 6: Acceleration evaluation of Hourglass-based Sparse Stereo Refinement.

| Method | Inference Speed(CPU) | Parameters |
|---|---|---|
| Hourglass-based SS Refinement (Proposed) | **1.41**s | **0.24**M |
| Parametric Skip Function (Liu et al. (2019)) | 4.22s | 8.64M |

In comparison to the baseline, our approach exhibits a significant speed improvement, being only 1/4 of the baseline's runtime. Additionally, our method achieves a parameter count that is merely 1/40 of the baseline's parameter count.

