# OpenReview forum: "FFCA-Net: Stereo Image Compression via Fast Cascade Alignment of Side Information"
_ICLR.cc/2024/Conference — Submitted to ICLR 2024_

### Official Review · Reviewer_fJJD · 2023-10-30

**Soundness:** 3 good
**Presentation:** 3 good
**Contribution:** 2 fair
**Rating:** 5
**Confidence:** 3

**Summary:**

The paper introduces the Feature-based Fast Cascade Alignment network (FFCA-Net), a novel approach to stereo image compression (SIC) suited for applications such as 3D imaging and car-mounted cameras. Contrary to conventional methods reliant on joint encoding and decoding, which place a substantial burden on the encoding terminal, FFCA-Net adopts a strategy of independent encoding and joint decoding, effectively minimizing decoding latency. The network implements a coarse-to-fine cascaded alignment methodology. Initially, it applies a feature domain patch-matching module to reduce redundancy and noise, succeeded by a sparse stereo refinement network aimed at precise alignment of inter-image features. Subsequently, a lightweight yet potent Fast Feature Fusion network (FFF) is utilized to adeptly decode the aligned features.

**Strengths:**

- The proposed FFCA-Net showcases exceptional performance, outshining competing methods, including those from recent publications, in various evaluations.
- The innovative modules introduced in the paper contribute significantly to enhancing image quality, as evidenced by the results presented in Table 3.

**Weaknesses:**

- The foundational concepts of the paper, including the coarse-to-fine cascade structure, hourglass-like stereo rectification, and feature fusion layer, appear to be somewhat straightforward, potentially raising questions about the novelty of the approach.
- The paper would benefit from a detailed discussion on the limitations and potential challenges associated with the proposed FFCA-Net.
- Typos and Formatting:
    - On Page 3, change “simultaneously” to “Simultaneously.”
    - Ensure consistent punctuation at the end of equations (1-6, 8-9) with periods, and a comma at the end of equation (7).
- For enhanced clarity and consistency in presentation, ensure that all numerical values across tables, such as Table 2 and Table 3, are expressed with the same number of decimal places.

**Questions:**

- Are there any limitations to the proposed method?

---

> ### Author Response · Authors · 2023-11-17
> **Response to Reviewer fJJD**
>
> We sincerely appreciate the time you have dedicated and the constructive recommendations you have provided. Your enthusiastic recognition of the originality of our article, the soundness of its structure, and the effectiveness of its experimental outcomes is a great source of inspiration for us! We are dedicated to addressing your inquiries and alleviating any concerns you may have.
>
> **Q1**: Are there any limitations to the proposed method?
>
> **R1:** Thank you for your inquiry, and we deeply apologize for not providing a clear description of the limitations of our work in the submitted version. Although our work presents an innovative lightweight framework that achieves state-of-the-art compression results for stereo images, it still has some inherent shortcomings, which we will now explain.
>
> - Our framework integrates a significant amount of prior knowledge about stereo images, including the presence of only horizontal displacements between image pairs. This allows our decoder to quickly and effectively combine side-information, leading to satisfactory decoding performance. However, our method relies solely on the prior information of specific scene image pairs. For general multi-view scenes, how to effectively utilize prior information from related viewpoints to assist compression is a future research question worth exploring. At the same time, our method currently only considers two related sources of information, so the expansion to multiple viewpoints $\geq3$) is also a worthwhile exploration.
>
>
> - Furthermore, while our proposed solution exhibits significantly faster decoding speed compared to learning-based SIC compression models of the same type, there still exists a gap in decoding speed when compared to non-learning-based traditional compression approaches (e.g. Jpeg). This implies that our method is still some distance away from being practically deployed on mobile devices.
>
>
> Thank you once again for your kind suggestion. We will include a description of the limitations of our work in the revised version.
>
>
>
> We have taken note of your astute observations regarding the weaknesses in our work, and we have carefully considered your concerns. Please allow us to provide an explanation for these inquiries.
>
> **W1**: The foundational concepts of the paper, including the coarse-to-fine cascade structure, hourglass-like stereo rectification, and feature fusion layer, appear to be somewhat straightforward, potentially raising questions about the novelty of the approach.
>
> **Rw1**: We truly understand your concerns, but respectfully, we disagree with that assertion. To the best of our knowledge, the coarse-to-fine cascade architecture we propose, along with its constituent modules, represents a novel application in the field of image compression. Compared to previous works, our approach offers a more rational integration of priors and a lighter, faster feature-matching architecture.
>
> - For example, in [1], there was an attempt to use stereo-matching techniques for feature domain matching. However, the disparity estimation module was found to be overly complex, leading to significant differences in inference speed compared to our proposed solution (relevant experimental results can be found in Section 4.2).
> - In addition, it is extremely difficult to adapt many established stereo matching techniques to the feature domain. For example, GC-Net [1] requires about 0.9 seconds to infer a single image on KITTI using a GPU [2]. If we were to adapt it to the feature domain, the computational cost would significantly increase due to the considerable computational overhead introduced by multiple channels. In contrast, our proposed "hourglass-based sparse stereo refinement" achieves an inference time of just 0.02 seconds on KITTI.
>
> - Unlike complex parameterized neural networks, our cascaded structure enables model-level editing, such as replacing coarse or fine alignment modules, making it convenient for future work.
>
>
> [1]Kendall A, Martirosyan H, Dasgupta S, et al. End-to-end learning of geometry and context for deep stereo regression[C]//Proceedings of the IEEE international conference on computer vision. 2017: 66-75.
>
> [2]Tankovich V, Hane C, Zhang Y, et al. Hitnet: Hierarchical iterative tile refinement network for real-time stereo matching[C]//Proceedings of the IEEE/CVF Conference on Computer Vision and Pattern Recognition. 2021: 14362-14372.
>
>
> **W2**: Typos and formatting error
>
> **Rw1**: We sincerely appreciate your valuable feedback on our work and the constructive suggestions you have provided. We apologize for any misunderstandings caused by printing errors. In the revised version, we have corrected these errors and thoroughly reviewed the entire article to prevent similar mistakes from happening again. We hope this time it will offer you a better reading experience!

---

> ### Author Response · Authors · 2023-11-20
> **Thanks to Reviewer fJJD**
>
> We sincerely appreciate your review and valuable feedback. Your positive evaluation of the comprehensiveness of our work and its outstanding performance is truly inspiring to us.
>
> We hope that our response has further solidified your recognition of the innovation in our work. If you have any remaining doubts or concerns, we are more than willing to alleviate them before the rebuttal period concludes.

---

> ### Author Response · Authors · 2023-11-21
> **A Gentle Reminder of the Final Feedback**
>
> We deeply value your positive feedback and the profound insights you have shared. We acknowledge that you may be currently occupied with various commitments and obligations. Nevertheless, we kindly request a moment of your time to review our responses to your concerns. We genuinely appreciate any further feedback you can provide. Additionally, we kindly ask for your consideration in updating the rating if we have effectively addressed your inquiries. We remain at your disposal to address any additional questions you may have before the conclusion of the rebuttal period.

---

> ### Author Response · Authors · 2023-11-22
> **A Second Reminder of the Post-rebuttal Feedback**
>
> Dear Reviewer fJJD,
>
> We greatly appreciate your initial comments. We totally understand that you may be extremely busy at this time. But we still hope that you could have a quick look at our responses to your concerns. We appreciate any feedback you could give to us. We also hope that you could kindly update the rating if your questions have been addressed. We are also happy to answer any additional questions before the rebuttal ends.
>
> Best Regards,
>
> Paper4896 Authors

---

> ### Comment · Reviewer_fJJD · 2023-11-22
> **Thank you for the response**
>
> We appreciate the authors' responses to my inquiries. I maintain my initial assessment, rating the submission as 'marginally below the acceptance threshold'.
>
> As a reviewer of the paper concerning image compression using the coarse-to-fine cascade architecture, I acknowledge the method's strengths in computational efficiency and performance. The adaptation of the coarse-to-fine cascade architecture, a well-established technique in computer vision [T1-T3], to the field of image compression, is indeed innovative in its application. However, I hold reservations about the novelty of the method due to its foundational reliance on previously established network models. The authors' stated points of novelty, such as significant differences in inference speed and the achievement of a 0.02-second inference time on the KITTI dataset, primarily focus on performance outcomes rather than on technical or conceptual breakthroughs.
>
> In several areas of computer vision, such as segmentation, video instance/object segmentation, and image super-resolution, the incorporation of architectures like the Swin-Transformer has led to substantial performance enhancements. Despite this, similar adaptations are often not regarded as groundbreaking enough for publication in major CV/ML conferences like ICLR or CVPR.
>
> Novelty is, admittedly, a subjective criterion in peer review. While some may argue that applying the coarse-to-fine cascade architecture in a new domain is novel, others might disagree. In my assessment, the technique, while effective, does not offer sufficient technical novelty or distinctiveness for acceptance in top-tier conferences. The points of novelty highlighted by the authors primarily pertain to the results rather than to the technical or conceptual innovations of the method.
>
> While the proposed method demonstrates impressive performance and efficiency, these aspects alone do not fully compensate for the lack of novelty in its technical approach. This dilemma has led me to a borderline decision between acceptance and rejection. The high performance of the proposed method is commendable and merits consideration, yet the question of novelty remains a pivotal factor in my final assessment.
>
> [T1] Wang, Zhenyang, Zhidong Deng, and Shiyao Wang. "CasNet: A cascade coarse-to-fine network for semantic segmentation." Tsinghua Science and Technology 24.2 (2018): 207-215.
>
> [T2] Cao, Yan-Pei, et al. "Learning to reconstruct high-quality 3D shapes with cascaded fully convolutional networks." Proceedings of the European Conference on Computer Vision (ECCV). 2018.
>
> [T3] Sahbi, Hichem. "Cascaded Coarse-to-Fine Deep Kernel Networks for Efficient Satellite Image Change Detection." arXiv preprint arXiv:1812.09119 (2018).

---

> > ### Author Response · Authors · 2023-11-22
> > **Thank You for Your Feedback**
> >
> > We appreciate your response. We have noticed that you still have some concerns regarding the novelty of our work. We are more than willing to provide experimental explanations in response to your *specific inquiries*. If you believe there are still areas where our paper falls short, we are more than happy to make further revisions to the revised version.

---

### Official Review · Reviewer_sqk4 · 2023-10-30

**Soundness:** 3 good
**Presentation:** 3 good
**Contribution:** 3 good
**Rating:** 6
**Confidence:** 3

**Summary:**

This paper presents FFCA-Net for the task of stereo image compression. Based on the fact in data compression, this method adopts independent encoding and joint decoding structure, and leverage the side information of decoder, proposing FFCA net based on the coarse-to-fine cascaded alignment.  With stereo patch matching, a pyramid-based sparse stereo refinement network, and a lightweight feature fusion module, the proposed method achieves good performance and fast decoding speed when compared with existing learning-based approaches.

**Strengths:**

The overall design is reasonable, and the realization is practical. Ablation study is provided to verify the effectiveness of the proposed technique components.

**Weaknesses:**

Most of the technique, although correct, seems incremental, either based on well-known fact/priors, or quite similar to existing method. For example, the fast fusion module, is similar to multi-frame fusion based works. However, other than this, the work is well-motivated, and reasonable designed to achieve good performances with fast speed.

**Questions:**

In Table 1, although most of the results are good, some of the best results belong to the method LDMIC.

Although not the same, some of the design componments are similar to the Huang 2023.

Fig. 2 needs to be refined. Current version, the text cannot be read clearly when printed.

---

> ### Author Response · Authors · 2023-11-17
> **Response to Reviewer sqk4**
>
> Dear Reviewer sqk4, we express our heartfelt gratitude for your gracious appreciation and insightful suggestions regarding our work. Your positive evaluation of the rationality of our model structure and the thorough reliability of our ablation experiments has truly inspired us. Moving forward, we shall address your inquiries in an endeavor to assuage any concerns you may have.
>
> **Q1**: In Table 1, although most of the results are good, some of the best results belong to the method LDMIC.
>
> Thank you for your insightful comments. LDMIC [1] is indeed a solid work that employs intricate cross-attention mechanisms to capture contextual information between image pairs. Nevertheless, I will illustrate why our approach holds greater potential in stereo image compression tasks through the following two points.
>
> - It's important to note that LDMIC optimizes its algorithm using Mean Squared Error (MSE) as the reconstruction loss, giving it an advantage in terms of the PSNR metric. In contrast, our proposed FFCA uses MS-SSIM as the loss function. From the results in Section 4.2 of our submission, it's evident that our method outperforms all baseline models in terms of MS-SSIM on all three datasets. Additionally, in some datasets, it also shows better performance in terms of PSNR compared to LDMIC.
> - Furthermore, our method strikes a more reasonable trade-off between compression performance and decoding speed. As seen in Section 4.2 in our submission, our proposed FFCA demonstrates significantly faster decoding speed compared to LDMIC (27.77s/4.92s).
>
>
> [1] Zhang X, Shao J, Zhang J. LDMIC: Learning-based Distributed Multi-view Image Coding[C]//The Eleventh International Conference on Learning Representations. 2022.
>
> **Q2**: Although not the same, some of the design components are similar to the Huang 2023.
>
> **R2**: Thank you for your insightful comment. Indeed, the work of reference [1] （Huang 2023）has provided us with a wealth of inspiration. However, it is essential to note that our proposed method deviates significantly from [1] in terms of both its structure and techniques.
>
> - **From a structural perspective**, [1] introduces a decoder that solely incorporates coarse-grained matching. In contrast, our proposed FFCA adopts a cascaded structure that operates in a coarse-to-fine manner. This allows our model to achieve more precise alignment in the feature space, reducing the noise introduced by coarse matching. Additionally, by integrating prior knowledge of stereoscopic imagery, we can avoid many of the artifacts associated with coarse matching. Our experimental results, as presented in the Section 4.2 , show that our architecture outperforms [1] in terms of Bit-rate by approximately 36.84% on the Instero2K dataset.
>
> - **From a technical standpoint,** our approach diverges from the patch-matching method proposed in [1], which relies on greedy search. Instead, we introduce stereo patch matching, enabling parallel search for similar blocks of features between perspectives. This results in several-fold improvements in matching speed. The table below presents a comparison of the matching speeds between the two methods.
>
>   |  Input Size$\downarrow$   | Method$\downarrow$, Speed$\rightarrow$  |     CPU     |    GPU     |
>   | :-----------------------: | :-------------------------------------: | :---------: | :--------: |
>   | 128$\times$256$\times$256 |    Stereo Patch-Matching (Proposed)     | **218.3ms** | **11.8ms** |
>   | 128$\times$256$\times$256 | Multi-scale Patch-Matching (Huang 2023) |  2163.3ms   |  138.8ms   |
>
>   The evident speed improvement of our proposed solution can be attributed to the novel techniques employed in its algorithmic implementation. While the work presented in [1] is solid, its high complexity poses challenges for further practical applications. In contrast, our work takes a significant stride toward practical deployment.
>
> [1] Huang Y, Chen B, Qin S, et al. Learned distributed image compression with multi-scale patch matching in feature domain[C]//Proceedings of the AAAI Conference on Artificial Intelligence. 2023, 37(4): 4322-4329.

---

> ### Author Response · Authors · 2023-11-17
> **Author Response (Part II)**
>
> **Q3**: Fig. 2 needs to be refined. Current version, the text cannot be read clearly when printed.
>
> **R3:** Thank you for your insightful comment! We apologize for the confusion caused by the unclear illustration. In the revised version, we have redrawn Fig. 2, emphasized the font in the image. You can see it in the newly submitted version.
>
>
>
> Thank you for raising your concerns in the weaknesses section. We have carefully reviewed and summarized your worries. Please allow us to provide appropriate explanations for these concerns.
>
> **W1**：Most of the technique, although correct, seems incremental, either based on well-known priors, or similar to the existing method.
>
> **Rw1**：We appreciate your comment! However, we respectfully disagree with this viewpoint. The priors of stereo images are well-known, but applying them effectively in a compression model requires skill.
>
> - Previous works in the SIC framework [1] [2] have attempted to combine these priors to improve compression performance. However, these approaches have not led to satisfactory performance improvements. They mainly concentrate on implicitly learning these priors through parameterized neural networks, which often results in slower inference speed. In contrast, we have efficiently integrated the priors of stereo images, achieving significant performance enhancements. This is, to the best of our knowledge, the first application of stereo matching techniques within the DSC framework.
> - While stereo matching is a well-established research field, the challenge lies in transferring such complex matching techniques from the image domain to the feature domain. Our experiments have shown that per-pixel learning of disparities in the feature domain introduces significant computational complexity that is hard to manage. Moreover, unlike matching in the image domain, feature domain stereo matching is more susceptible to introducing noise, making training more demanding.
>
> In conclusion, we think that our proposed coarse-to-fine feature domain matching strategy is insightful and can provide new directions for future learning-based DSC algorithms.
>
> [1] Liu J, Wang S, Urtasun R. Dsic: Deep stereo image compression[C]//Proceedings of the IEEE/CVF International Conference on Computer Vision. 2019: 3136-3145.
>
> [2] Deng X, Yang W, Yang R, et al. Deep homography for efficient stereo image compression[C]//Proceedings of the IEEE/CVF Conference on Computer Vision and Pattern Recognition. 2021: 1492-1501.

---

> ### Author Response · Authors · 2023-11-20
> **Thanks to Reviewer sqk4**
>
> Please allow us to thank you again for reviewing our paper and the valuable comments, particularly for your recognition of our rational and innovative model architecture, as well as the comprehensive nature of our experiments.
>
> We kindly request your feedback regarding whether our response and the new experiments have adequately addressed your concerns. We are delighted to address any further inquiries during the post-rebuttal phase. Your input is highly valued and greatly appreciated.

---

> ### Author Response · Authors · 2023-11-21
> **A Gentle Reminder of the Final Feedback**
>
> We extend our heartfelt appreciation for your initial feedback, which holds immense value in enhancing our work.  We would be grateful if you could take a moment to review our response and modifications, and kindly let us know if anything else that can be added to our next version.

---

> ### Author Response · Authors · 2023-11-22
> **A Second Reminder of the Post-rebuttal Feedback**
>
> Dear Reviewer sqk4,
>
> We highly appreciate your initial feedback and acknowledge the possibility of your busy schedule. Nevertheless, we humbly request your brief attention to our responses addressing your concerns. Any feedback you can provide would be immensely appreciated. Moreover, if your inquiries have been adequately addressed, we would be thankful if you could kindly consider adjusting your rating accordingly. Additionally, we remain at your disposal to address any further questions before the conclusion of the rebuttal period.
>
> Best Regards,
>
> Paper4896 Authors

---

> > ### Comment · Reviewer_sqk4 · 2023-11-22
> > **Thanks for your response**
> >
> > Thanks for your efforts in addressing my questions. I maintain my original score, " marginally above the acceptance threshold"

---

> > > ### Author Response · Authors · 2023-11-22
> > > **Thank You for Your Feedback**
> > >
> > > Your evaluation fills us with profound inspiration, and for that, we express our heartfelt gratitude.

---

### Official Review · Reviewer_YULU · 2023-10-31

**Soundness:** 3 good
**Presentation:** 2 fair
**Contribution:** 3 good
**Rating:** 6
**Confidence:** 4

**Summary:**

This manuscript utilizes features and prior knowledge of stereo images in their proposed stereo image compression framework, which achieves advanced compression performance and accelerates decoding time by 3-10 times compared to other learning based methods.

**Strengths:**

Originality: This paper uses the prior knowledge of stereo images for decoding acceleration, resulting in strong originality.

Quality: This paper provides a detailed description of the prior assumptions used, which is reasonable to some extent; This paper has conducted sufficient experiments to prove that the proposed method can achieve the most advanced compression performance and faster compression speed.

Clarity: This paper provides a clear introduction to the background and motivation of stereo image compression through an abstract, as well as the improvements and advantages of the proposed method compared to other compression algorithms; The structure of the paper is complete and the overall description is relatively clear.

Importance: This paper should have certain reference value for the field of stereo image compression. The proposed method can maintain the compression performance at the STOA level and achieve 3-10 times faster decoding speed than previous learning based algorithms.

**Weaknesses:**

The improvement of network encoding speed in this manuscript mainly relies on prior knowledge of binocular stereo images. However, these prior assumptions may not be applicable in all cases. Overreliance on prior knowledge may result in the loss of information, which the neural network itself can learn that is beneficial for encoding reconstruction.

The description of techniqe details are not clear or complete enough. For example, , what is the relationship between main and side image in the method section? For non overlay patches, which kind of operation should be done if there are non integer patches remaining. The description of the dataset in the experimental section was too simplistic, such as not introducing scale, resolution, scene, etc.

Although the decoding speed of the method proposed in this paper is significantly improved compared to other methods, the average decoding time for images with a resolution of 832x1024 is 4.91 seconds, which is still difficult to accept in practical applications.

**Questions:**

1. What is the relationship between Main image and Side image? Is the side image encoded? How to encode? How to measure the final decoding time?

2. Which module mentioned in this paper has the most significant acceleration on image decoding?

3. Do the boundary patches at both ends of the binocular stereo image in the dataset used in the paper require special processing operations?

4. In the definition of G in formula (7), if the distance between two features is less than a certain threshold, it is actually a significant difference. Is there a problem with the description here? What is the definition of $G^c$?

5. Does the bpp term in the loss function only contain the potential representation z, without using any prior knowledge from other learning based compression methods? Is it already included? Are there any special considerations for model optimization based on MS-SSIM?

6. In the ablation experiment, the Fast Feature Fusion module slightly reduced PSNR. Which result proves that the FFF module can achieve faster decoding? How many iterations can the FFF module undergo to achieve the best results?

---

> ### Author Response · Authors · 2023-11-17
> **Response to Reviewer YULU**
>
> Dear Reviewer YULU, we sincerely express our gratitude for your invaluable time and constructive suggestions. Your positive appraisal of our article's originality, rational framework, and completeness experimental results is truly inspiring! Moving forward, we will address your inquiries and alleviate your concerns.
>
> **Q1**: What is the relationship between Main image and Side image? Is the side image encoded? How to encode? How to measure the final decoding time?
>
> **R1**: Thank you sincerely for your feedback! We deeply apologize for any confusion our description may have caused you. Please allow me to provide further clarification.
>
> - In the information theory [1], the terms "side/main information" are proprietary terminology within the DSC framework, which proves that compression of two correlated sources that do not communicate with each other could achieve compression efficiency with mutual communication. Since our work also falls within the DSC framework, we have used the same terminology [2] [3] in our work, **referring to different perspectives in stereoscopic imagery as "main/side image."**
> - **The side image is also compressed**. In practical applications, the concept of "main/side image" is relative. Users can choose their current perspective as the main image and the others as the side image for auxiliary decoding, and vice versa. This means that after another round of encoding and decoding, the contents of the main and side images can be swapped, and the difference can be adjusted to decode the other perspective.
> -  Since this process is symmetric, the side image is encoded as the main one, independently. **the average decoding time for both perspectives is theoretically equivalent to decoding a single image**. Hence, in our experiments, we simulate the decoding time for a pair of stereo images by using twice the time it takes to decode a single image.
>
> Thank you once again for your feedback. We acknowledge that our previous submission lacked clarity in explaining this issue. We will make sure to provide additional clarification in the revised version.
>
>
>
> [1] Cover, T. 1975. A proof of the data compression theorem of Slepian and Wolf for ergodic sources (Corresp.). *IEEE* *Transactions on Information Theory*, 21(2): 226–228.
>
> [2] Huang Y, Chen B, Qin S, et al. Learned distributed image compression with multi-scale patch matching in feature domain[C]//Proceedings of the AAAI Conference on Artificial Intelligence. 2023, 37(4): 4322-4329.
>
> [3] Ayzik S, Avidan S. Deep image compression using decoder side information[C]//Computer Vision–ECCV 2020: 16th European Conference, Glasgow, UK, August 23–28, 2020, Proceedings, Part XVII 16. Springer International Publishing, 2020: 699-714.

---

> ### Author Response · Authors · 2023-11-17
> **Author Response (Part II)**
>
> **Q2:** Which module mentioned in this paper has the most significant acceleration on image decoding?
>
> **R2:** We apologize for the lack of comprehensive experimental results on acceleration in the initial submission. Please allow me to present the following ablation result to address your concerns.
>
> We have divided our proposed solution into three parts: coarse-grained stereo patch matching, fine-grained hourglass-based sparse stereo refinement, and feature fusion using the FFF module. To ensure fairness, we used MSFDPM [1] as our baseline to show the most significant module for acceleration on image decoding.
>
> - **Stereo Patch Matching**:  We have selected Multi-Scale Patch Matching [1] as the baseline method for comparison. Both approaches were evaluated on a feature layer of size $128\times256\times256$.
>
>   |  Input Size$\downarrow$   | Method$\downarrow$, Speed$\rightarrow$ |     CPU     |    GPU     |
>   | :-----------------------: | :------------------------------------: | :---------: | :--------: |
>   | 128$\times$256$\times$256 |    Stereo Patch-Matching (Proposed)    | **218.3ms** | **11.8ms** |
>   | 128$\times$256$\times$256 |  Multi-scale Patch-Matching (MSFDPM )  |  2163.3ms   |  138.8ms   |
>
>   ​	It is evident from the results that our algorithm outperforms the baseline method by nearly tenfold, both in CPU and GPU environments. This   significant speed improvement is attributed to our efficient parallel computing techniques, which have proven to be reliable.
>
> - **Fast Feature Fusion**: We selected the Feature Fusion module [1] as the baseline for comparison. Our proposed FFF  module is an enhanced version of the Feature Fusion module, with a smaller parameter count and faster inference speed. We evaluated the inference speed on CPU and the model parameter count for both methods, with an input size of $3\times512\times512$, as shown in the table below. The results confirm the effectiveness of our model.
>
>   | Input Size$\downarrow$  | Method$\downarrow$, Efficiency$\rightarrow$ | Speed(CPU)  | Parameters |
>   | :---------------------: | :-----------------------------------------: | :---------: | :--------: |
>   | 3$\times$512$\times$512 |       Fast Feature Fusion (Proposed)        | **577.5ms** | **3.04M**  |
>   | 3$\times$512$\times$512 |           Feature Fusion (MSFDPM)           |   683.9ms   |   7.02M    |
>
>
>
> Based on our ablation experiments and in comparison to the baseline MSDFPM, **the most significant speed improvement is observed in our proposed stereo patch matching technique.** This high-speed and robust coarse-grained alignment provides a solid foundation for subsequent decoding processes.
>
> Furthermore, our proposed module 'Hourglass-based Sparse Stereo Refinement' performs fine-grained adjustment on top of coarse-grained matching, which is not present in MSFDPM. However, even with the addition of pixel-level correction, our method still exhibits a significant speed advantage. As shown in Section 4.2 of the original paper, the decoding speed of our approach is 1/5 that of MSFDPM (4.9s/23.85s).
>
> As far as we know, previous works utilizing the DSC framework did not integrate stereo matching techniques. To further confirm the contribution of our fine-grained module in terms of acceleration, we chose the Parametric Skip Function [2], a crucial component of a non-DSC stereo image compression algorithm (DSIC), as the baseline method. This method aims to predict complete inter-feature disparities using stacked residual blocks. We assessed the inference speed on CPU and the model parameter count for both methods, with an input size of $3\times512\times512$, as indicated in the table below:
>
> | Input Size$\downarrow$  | Method$\downarrow$, Efficiency$\rightarrow$ | Speed(CPU) | Parameters |
> | :---------------------: | :-----------------------------------------: | :--------: | :--------: |
> | 3$\times$512$\times$512 |  Hourglass-based SS Refinement (Proposed)   | **308ms**  | **0.24M**  |
> | 3$\times$512$\times$512 |       Parametric Skip Fuction (DSIC)        |  1194.3ms  |   8.64M    |
>
> In comparison to the baseline, our approach exhibits a significant speed improvement, being only 1/4 of the baseline's runtime. Additionally, our method achieves a parameter count that is merely 1/40 of the baseline's parameter count.
>
> We express our gratitude once again for your insightful commentary. We shall incorporate these experiments into the appendix, and you will find them in Section 6.2 of the revised version.
>
> [1] Huang Y, Chen B, Qin S, et al. Learned distributed image compression with multi-scale patch matching in feature domain[C]//Proceedings of the AAAI Conference on Artificial Intelligence. 2023, 37(4): 4322-4329.
>
> [2] Liu J, Wang S, Urtasun R. Dsic: Deep stereo image compression[C]//Proceedings of the IEEE/CVF International Conference on Computer Vision. 2019: 3136-3145.

---

> ### Author Response · Authors · 2023-11-17
> **Author Response (Part III)**
>
> **Q3:** Do the boundary patches at both ends of the binocular stereo image in the dataset used in the paper require special processing operations?
>
> **R3**: Thank you for your comments. We apologize for any confusion caused by the oversight. In practical applications, we will use replication padding on the feature map of the main image ($h^i_\hat{x}$) to ensure that they can be evenly divided by the patch size $B$. This is a commonly used technique [1-2] and has its rationale: due to the small size of the patches we used, the replication-padding approach not only maintains intra-block similarity but also ensures consistency between the padded boundaries and the original boundaries. Once the matching is completed, we simply trim the matched results back to their original size. We appreciate your suggestion, and we will include these processing steps in the revised appendix as part of the experimental setup.
>
>
> [1] Huang Y, Chen B, Qin S, et al. Learned distributed image compression with multi-scale patch matching in feature domain[C]//Proceedings of the AAAI Conference on Artificial Intelligence. 2023, 37(4): 4322-4329.
>
> [2] Ayzik S, Avidan S. Deep image compression using decoder side information[C]//Computer Vision–ECCV 2020: 16th European Conference, Glasgow, UK, August 23–28, 2020, Proceedings, Part XVII 16. Springer International Publishing, 2020: 699-714.
>
>
>
> **Q4**: In the definition of  $G$  in formula (7), if the distance between two features is less than a certain threshold, it is actually a significant difference. Is there a problem with the description here? What is the definition of $G^c$?
>
> **R4:** Thank you for your insightful suggestion! We apologize for the print error that affected your reading. Actually, it should be a “$\geq$” symbol, indicating that all feature channels exceeding a certain threshold will be selected as the set $G$.
>
> Meanwhile, $G^c $represents the complement set of $G$, defining the other channels that are below the given threshold. To facilitate better understanding for readers, we have corrected this print error and provided the corresponding explanation for $G^c $ in the revised version.

---

> ### Author Response · Authors · 2023-11-17
> **Author Response (Part IV)**
>
> **Q5**:  Does the bpp term in the loss function only contain the potential representation z, without using any prior knowledge from other learning-based compression methods? Is it already included? Are there any special considerations for model optimization based on MS-SSIM?
>
> **R5**: Thank you for posing this inquiry. We regret that we were unable to articulate this matter clearly in the submitted version.
>
> **Regarding the 'Does the bpp term in the loss function only contain the potential representation z?',** the bpp term typically encompasses prior knowledge and is determined by the components of the baseline single-image encoder-decoder we have employed. In this manuscript, we utilized [1] as the baseline single-image compression model, thereby incorporating the mixture Gaussian hyperprior offered in [1] within the bpp term, as depicted in the following equation.
> $$
> \begin{aligned}
> \mathcal{R}({\boldsymbol{z}})= \mathcal{R}(\hat{\boldsymbol{z}})+\mathcal{R}(\hat{\boldsymbol{u}})
> \end{aligned}
> $$
>
> $$
> \mathcal{R}(\hat{\boldsymbol{z}})=\mathbb{E}\left[-\log \left(p_{\hat{\boldsymbol{z}} \mid \hat{\boldsymbol{u}}}(\hat{\boldsymbol{z}} \mid \hat{\boldsymbol{u}})\right)\right]
> $$
>
> $$
> \mathcal{R}(\hat{\boldsymbol{u}}) =\mathbb{E}\left[-\log \left(p_{\hat{\boldsymbol{u}} \mid \boldsymbol{\psi}}(\hat{\boldsymbol{u}} \mid \boldsymbol{\psi})\right)\right]
> $$
>
> $$
> \text{where} \quad p_{\hat{\boldsymbol{z}} \mid \hat{\boldsymbol{u}}}(\hat{\boldsymbol{z}} \mid \hat{\boldsymbol{u}}) \sim \sum_{k=1}^K \boldsymbol{w}^{(k)} \mathcal{N}\left(\boldsymbol{\mu}^{(k)}, \boldsymbol{\sigma}^{2(k)}\right)
> $$
>
> To provide readers with a better understanding of this aspect, we will include these explanatory details in the revised appendix.
>
> **Regarding the " Are there any special considerations for model optimization based on MS-SSIM?",** In practice, it is common to optimize using MS-SSIM as a reconstruction loss. Although this may lead to a slight decrease in PSNR during testing, we have found in our experiments that using MSE as the reconstruction loss can sometimes produce unsatisfactory results. For example, at low bit rates, models trained with MSE loss (such as BPG) may have acceptable PSNR but often suffer from distortions and artifacts around object edges (this point is also illustrated in the original submission through visualizations). On the other hand, using MS-SSIM helps address these issues. We believe that image tearing and distortions are more unacceptable than a slightly blurred visual effect, which is a significant reason for choosing MS-SSIM as the reconstruction loss.
>
>
>
> [1] Cheng Z, Sun H, Takeuchi M, et al. Learned image compression with discretized gaussian mixture likelihoods and attention modules[C]//Proceedings of the IEEE/CVF conference on computer vision and pattern recognition. 2020: 7939-7948.
>
> [2] Huang Y, Chen B, Qin S, et al. Learned distributed image compression with multi-scale patch matching in feature domain[C]//Proceedings of the AAAI Conference on Artificial Intelligence. 2023, 37(4): 4322-4329.
>
> [3] Ayzik S, Avidan S. Deep image compression using decoder side information[C]//Computer Vision–ECCV 2020: 16th European Conference, Glasgow, UK, August 23–28, 2020, Proceedings, Part XVII 16. Springer International Publishing, 2020: 699-714.
>
> [4] Zhang X, Shao J, Zhang J. LDMIC: Learning-based Distributed Multi-view Image Coding[C]//The Eleventh International Conference on Learning Representations. 2022.

---

> ### Author Response · Authors · 2023-11-17
> **Author Response (Part V)**
>
> **Q6:** In the ablation experiment, the Fast Feature Fusion module slightly reduced PSNR. Which result proves that the FFF module can achieve faster decoding? How many iterations can the FFF module undergo to achieve the best results?
>
> **R6**: We appreciate your constructive comments! Please allow me to address each of your questions separately.
>
> - **Which result proves that the FFF module can achieve faster decoding?**  R:  Thank you for pointing out this aspect. You can find the additional experimental results in Table of Respond 2. The ablation experiments demonstrate that our FFF module is more lightweight (3.04M/7.02M) and exhibits improved inference speed (577.5ms/683.9ms). We have also included the corresponding experimental results in the revised version, which can be found in Section 6.2.
>
> - **How many iterations can the FFF module undergo to achieve the best results?** R: We apologize for any misunderstanding caused by the omission in the description. The number of iterations in the FFF module is indeed linked to the number of scales of the multi-scale features separated by the baseline encoder. For instance, in this work, we employed a baseline encoder-decoder that decouples the image into four distinct scale feature layers, denoted as .$h^1_\hat{x},h^2_\hat{x},h^3_\hat{x},h^4_\hat{x}$, arranged in decending order of scale. Each iteration of the FFF module can be abstracted by the following equation.
>   $$
>   \phi^i=FFF^i(\phi^{i+1},h^i_\hat{x},h^i_{y^{\star\star}}) \quad i=1,2,3,4
>   $$
>   Here, $FFF^i$ refers to the FFF module at the i-th iteration, and $h^i_{y^{\star\star}}$ represents the side information obtained through coarse to fine matching. As observed, it takes **4 iterations** of FFF to fuse information from all four feature scales.
>
> Additionally, omitting any scale of features in the decoding stage hinders achieving improved image reconstruction[1]. Here, we conduct an ablation study using the InStero2k dataset to illustrate this point. The variable in focus is the scale of features used by the decoder, with $\lambda=0.01$.
>
>   | The fused features | $h^{1,2,3,4}$ | $h^{1,2,3}$ | $h^{1,2}$ | $h^1$ |
>   | :----------------: | :---------------: | :-----------: | :-------: | :---: |
>   |      **PSNR**      |       32.8        |     32.1      |   30.3    | 29.2  |
>
> This aligns with our expectations, as the **4 iterations of FFF can effectively fuse features from all scales and achieve optimal reconstruction results.** Once again, we appreciate your feedback, and we will include these supplementary explanations in Section 6.2 of the revised version.

---

> ### Author Response · Authors · 2023-11-17
> **Author Response (Part VI)**
>
> Thank you for bringing up your concerns in the weaknesses section. We have thoroughly reviewed and addressed your worries. Please let us offer suitable explanations for these concerns.
>
> **W1**: The prior assumptions may not be applicable in all cases. Overreliance on prior knowledge may result in the loss of information, which the neural network itself can learn that is beneficial for encoding reconstruction.
>
> **Rw1**: Thank you for your constructive insight! Although the prior algorithm we provided may not be effective in certain non-stereo image compression scenarios, we believe that even in these situations, our work still offers several insightful aspects.
>
> -  Utilizing these non-learning priors can effectively accelerate the inference process of the model. Generally, employing complex parameterized neural networks to learn priors often leads to unsatisfactory encoding and decoding speeds.
> - While the priors may vary across different scenarios, our cascaded structure provides a valuable framework and solution for integrating this prior knowledge effectively.
>
> However, it is acknowledged that the applicability of our approach may be limited in certain scenarios. In our future works, we will strive to address and potentially alter this limitation.
>
> **W2**：The average decoding time for images is still difficult to accept in practical applications
>
> **Rw2**: Thank you for your thoughtful insights! Indeed, achieving satisfactory speeds with current learning-based deep compression algorithms, especially within the framework of stereo image compression, remains challenging. **Nevertheless, compared to other SIC methods, our approach has demonstrated noticeable improvements in decoding speed.** We firmly believe that by incorporating stronger priors and effectively integrating them, we can achieve fast compression algorithms that hold significant practical value, particularly in more specialized scenarios. In this sense, our method has taken a step forward in this research direction.
>
> **W3**: The description of technique details is not clear or complete enough.
>
> **Rw3**: Thank you for your response! We have addressed the first two questions in detail in our previous answers, which you can refer to. We apologize for the inconvenience caused by the limited length of the article, which prevented us from providing a detailed description of the dataset preprocessing process and the specifics of the dataset itself. To provide readers with a better understanding, we have included the details of the preprocessing and dataset in the revised version's appendix, which can be found in Section 6.1.

---

> ### Author Response · Authors · 2023-11-20
> **Thanks to Reviewer YULU**
>
> We express our gratitude once again for your review of our work and the valuable feedback provided, particularly regarding its strong originality, comprehensive experimental results, and good writing.
>
> We would be delighted to address any questions or concerns you may have regarding the experimental design details or the contributions of modules to acceleration before the rebuttal period concludes.

---

> ### Author Response · Authors · 2023-11-21
> **A Gentle Reminder of the Final Feedback**
>
> We express our utmost gratitude for your insightful and inspiring comments. We sincerely hope that our explanations regarding the *hyperparameter settings* and the *ablation studies on inference acceleration between modules* have effectively alleviated your concerns. Should you have any additional remarks, please do not hesitate to inform us, as we are more than willing to provide you with further clarification.

---

> ### Author Response · Authors · 2023-11-22
> **A Second Reminder of the Post-rebuttal Feedback**
>
> Dear Reviewer YULU,
>
> We deeply value your initial feedback and understand that you may have a busy schedule. However, we kindly request that you take a moment to review our responses to your concerns. Any feedback you can provide would be greatly appreciated. Furthermore, if your questions have been addressed, we would be grateful if you could consider updating your rating accordingly. We are also available to address any additional questions before the rebuttal period concludes.
>
> Best Regards,
>
> Paper4896 Authors

---

> > ### Author Response · Authors · 2023-11-23
> > **The Third Warm Reminder of the Post-rebuttal Feedback**
> >
> > Dear Reviewer YULU,
> >
> > We notice that all other reviewers have posted their post-rebuttal comments to our response but we still have not received any further information from you. We greatly appreciate your initial comments. We fully understand that you may be extremely busy at this time. However, we kindly request that you take a moment to review our responses addressing your concerns. Any feedback you can provide would be highly appreciated. Additionally, if your questions have been adequately addressed, we would be grateful if you could consider updating the rating accordingly.
> >
> > Best Regards,
> >
> > Paper 4896 Authors

---

### Author Response · Authors · 2023-11-17
**Modifications to the revision**

We express our gratitude to all the insightful remarks provided by the reviewers. Throughout the process of rebutting, we firmly believe that the concerns raised by the reviewers have been appropriately addressed. We hereby present a concise summary of the revised submission.

- We have rectified typos and formatting errors, including ensuring consistent decimal places for numerical results and correcting punctuation misuse.
- Section 3: We have revised Figure 2 to enhance its clarity, making the text on the figure more legible for readers.
- Section 3.2: We have added a description regarding channel sparsity to provide a clearer logical explanation. Furthermore, we have included definitions for certain mathematical symbols and corrected print error.
- Section 5: We have incorporated plans and prospects for future work in the conclusions.
- Appendix 6.1: We have provided additional details on the experimental setup, which includes a comprehensive description of the datasets used, the parameter settings for training on different datasets, and the preprocessing procedures employed for each dataset.
- Appendix 6.2: We have supplemented the ablation experiments within the model, focusing on the acceleration and lightweight characteristics of different modules.

We have **highlighted all non-typo changes in red font** to emphasize the modifications.

---

### Meta-Review · Area_Chair_f1CT · 2023-12-06

**Metareview:**

This paper propose a novel application of the feature cascade alignment for stereo image compression.
While the components of the proposed technique have been already developed in prior work, the new application shows their usefulness in stereo compression.

**Justification For Why Not Higher Score:**

Reviewers did not find the contributions of the work novel enough for publication in ICRL.

**Justification For Why Not Lower Score:**

N/A

---

### Decision · Program_Chairs · 2024-01-16

Reject